# Preparation, Physicochemical Properties, and In Vitro Toxicity towards Cancer Cells of Novel Types of Arsonoliposomes

**DOI:** 10.3390/pharmaceutics12040327

**Published:** 2020-04-06

**Authors:** Paraskevi Zagana, Spyridon Mourtas, Anastasia Basta, Sophia G. Antimisiaris

**Affiliations:** 1Foundation for Research and Technology Hellas, Institute of Chemical Engineering Sciences, FORTH/ICE-HT, 26504 Rio-Patras, Greece; voulazagana@yahoo.gr (P.Z.); mourtas@upatras.gr (S.M.); natasha_chem@hotmail.com (A.B.); 2Lab. Pharm. Technology, Department of Pharmacy, University of Patras, 26504 Rio-Patras, Greece

**Keywords:** arsenic, liposomes, doxorubicin, curcumin, anticancer, synergy, activity

## Abstract

Arsonoliposomes (ARSL) are liposomes that incorporate arsonolipids (ARS) in their membranes. They have demonstrated significant toxicity towards cancer cells, while being less toxic towards normal cells. In this study, we sought to investigate the possibility to prepare novel types of arsonoliposomes (ARSL) by incorporating a lipidic derivative of curcumin (TREG) in their membrane, and/or by loading the vesicles with doxorubicin (DOX). The final aim of our studies is to develop novel types of ARSL with improved pharmacokinetics/targeting potential and anticancer activity. TREG was incorporated in ARSL and their integrity during incubation in buffer and serum proteins was studied by monitoring calcein latency. After evaluation of TREG-ARSL stability, the potential to load DOX into ARSL and TREG-ARSL, using the active loading protocol, was studied. Loading was performed at two temperatures (40 °C and 60 °C) and different time periods of co-incubation (of empty vesicles with DOX). Calculation of DOX entrapment efficiency (%) was based on initial and final drug/lipid ratios. The cytotoxic activity of DOX-ARSL was tested towards B16F10 cells (mouse melanoma cells), LLC (Lewis Lung carcinoma cells), and HEK-293 (Human embryonic kidney cells). Results show that TREG-ARSL have slightly larger size but similar surface charge with ARSL and that they are both highly stable during storage at 4 °C for 56 d. Interestingly, the inclusion of TREG in ARSL conferred increased stability to the vesicles towards disruptive effects of serum proteins. The active-loading protocol succeeded to encapsulate high amounts of DOX into ARSL as well as TREG-LIP and TREG-ARSL, while the release profile of DOX from the novel liposome types was similar to that demonstrated by DOX-LIP. The cytotoxicity study results are particularly encouraging, since DOX-ARSL were less toxic towards the (normal) HEK cells compared to the two cancer cell-types. Furthermore, DOX-ARSL demonstrated lower toxicities (at all concentrations tested) for HEK cells, compared to that of the corresponding mixtures of free DOX and empty ARSL, while the opposite was true for the cancer cells (in most cases). The current results justify further in vivo exploitation of DOX-ARSL, as well as TREGARSL as anticancer therapeutic systems.

## 1. Introduction

Arsonolipids are analogues of phosphonolipids in which P has been replaced by as in the lipid head group. (Scheme 1) [1,2]. Arsonoliposomes (ARSL) constitute a particular class of liposomes, which are consisted of mixtures of arsonolipids, phospholipids, and cholesterol [2,3]. Nanosized ARSL demonstrate high toxicity against specific cancer cells, while being substantially less toxic towards normal cells, as proven by in vitro [4,5,6] and in vivo studies [7]. 

Specific ARSL compositions that include a polyethylene-glycol (PEG) coating on the lipid vesicle surface were demonstrated to have very high integrity during incubation in presence of serum proteins (compared to other ARSL compositions) [8,9], making them suitable for in vivo administration as cancer therapeutics [10,11]. In more detail, the in vitro integrity of ARSL composed of arsonolipid (C16), DSPC, and cholesterol (at 8:12:10 mole ratio) during incubation in serum proteins was highly increased when 8 mole% of DSPE-conjugated PEG was added in their membrane [9]. 

The in vitro integrity tests for ARSL formulations was proven to be a good predictive tool for the in vivo bioavailability of arsenic; ARSL types that demonstrated higher integrity were also found to have significantly improved pharmacokinetics (compare to the ARSL types with lower in vitro integrity) [10,11]. In more detail, the levels of arsenic measured in abdominal tissues and in blood were significantly higher after administration of the more stable (with higher in vitro integrity) DSPC-containing ARSL [11], compared to the less stable PC-containing ARSL [10]. Furthermore, when the ARSL were coated with PEG molecules (in the above mentioned lipid composition, which conferred highly stable vesicles [9]), the pharmacokinetic profile of arsenic was influenced even more, resulting in higher blood-circulation times, compared to those obtained by other (less stable) ARSL formulations [10,11]. 

Nevertheless, arsonoliposomes have some drawbacks, which if encountered successfully may increase their potential applicability. The two most important drawbacks are their low toxicity towards some cancer types, and their—up-to-date—inability to overcome the BBB. Indeed, while being highly toxic towards some cancer cell types, such as human leukemia cells (NB4 and HL60), prostatic cancer PC3 cells, and rat brain glioma C6 cells, ARSL did not exhibit the same high toxicity for other cancer cells, such as human breast adenocarcinoma MDA-MB-468 cells, and rat pituitary tumor GH3 cells [5,6]. Additionally, arsenic was not detected in the brain of animals injected with ARSL, in all the cases of ARSL which were studied in vivo [10,11], thus posing an interesting challenge for further exploitation. Indeed, if ARSL could overcome the blood brain barrier, they could potentially kill parasites in the central nervous system or act as anticancer therapeutics against difficult or even impossible to treat brain tumors.

Based on the identified most stable ARSL lipid composition, we attempted herein to investigate the potential to develop novel ARSL formulations that incorporate a curcumin lipidic derivative (TREG), as a method to increase their distribution in the brain; and/or encapsulate Doxorubicin (DOX), as a method to increase their toxicity towards cancer cells.

DOX is a highly potent anticancer drug; however, its serious side-effects, and especially its cardiotoxicity, restricts its use in many situations [12]. The latter problem has been resolved by the construction of PEGylated DOX-loaded liposomes, which realize altered DOX pharmacokinetics and consequently, reduced toxicity [12,13,14]. The easy loading of high amounts of DOX in pre-formed empty PEG-liposomes by an active-loading protocol [15,16] has highly contributed to the development of the long-circulating liposomal formulation of DOX, which is in clinical use for more than 25 years. 

Curcumin (CUR) is a polyphenol that demonstrates interesting anti-inflammatory activity, while its potential anticancer activity is a controversial issue [17]. A lipid derivative of curcumin, TREG, was previously synthesized and incorporated in liposomes, as an approach for development of nanoparticulate therapeutics for Alzheimer’s disease (AD) [18]. In addition to their high affinity for amyloid species [18,19], TREG-liposomes demonstrated increased brain targeting potential, while the integrity of TREG-incorporating liposomes during incubation in presence of serum proteins was increased, as demonstrated in the case of non-targeted as well as brain-targeted liposomes (compared to control liposomes without TREG, in both cases) [20,21]. 

For the reasons mentioned above, we investigated herein the ability to incorporate TREG in ARSL, as well as the potential to load ARSL-types with DOX. Since synergistic anticancer activity between CUR and DOX has been reported before [22,23], perhaps the presence of TREG on DOX-loaded ARSL may confer increased anticancer activity in addition to any potential increase of the vesicle stability and/or targeting capability. 

For the loading of DOX in both, ARSL and TREG-ARSL, the active loading procedure was applied [15,16]. Preliminary evaluation of the toxicity of some of the novel vesicle types towards cancer cell lines, and normal cells, was finally performed.

## 2. Materials and Methods

1,2-Distearoyl-sn-glycerol-3-phosphatidyl-choline [DSPC], and 1,2-Distearoyl-sn-glycerol-3-phosphatidyl-ethanolamine-*N*-[methoxy(polyethylene-glycol)-2000] [PEG2000] were purchased from Lipoid, Germany. Cholesterol (Chol) was purchased from Sigma-Aldrich (Darmstadt, Germany). The rac-2,3-dipalmitoyl-oxypropylarsonic acid [ARS] (C_16_), was synthesized as described in detail before [1,24,25]; TREG lipid was also synthesized using the method reported in detail before [18,19]. Doxorubicin, hydrochloric salt (DOX) was purchased by Tocris Bioscience, UK. Fetal Calf Serum (FCS) was from Sigma (Darmstadt, Germany).

A bath sonicator (Branson) and a microtip-probe sonicator (Sonics and Materials, Leicestershire, UK) were used for liposome preparation. Protein concentrations were measured by Bradford microassay (1–10 µg protein/mL). The Cell Viability MTT assay was carried out with 3-(4,5-Dimethylthiazol-2-yl)-2,5-diphenyltetrazolium bromide (MTT) reagent, which was purchased by Sigma-Aldrich. All other reagents and chemicals used were of analytical grade, and were purchased from Sigma-Aldrich. The osmolarity of solutions used for liposome formation was measured by a Roebling osmometer and adjusted to 300 mOsm with NaCl (if needed).

### 2.1. Preparation of TREG-Incorporating ARSL (TREG-ARSL)

Liposomes (ARSL and TREG-incorporating arsonoliposomes (TREG-ARSL)) with the lipid compositions reported in Table 1 were prepared as mentioned below. The specific ARSL composition used (DSPC/ARS/Chol (12:8:10 mol/mol containing 8 mole% PEG2000)) was the one that was previously identified to confer the highest integrity during incubation in presence of FCS (compared to other ARSL compositions) [9]. Additionally, conventional liposomes (without arsonolipids) (LIP) and TREG-incorporating liposomes (TREG-LIP) were also formulated to be used as control formulations. For all liposome types, the thin-film-hydration method followed by probe sonication for size reduction was applied [26]. In brief, lipids were dissolved in chloroform/methanol (2:1 *v*/*v*), and appropriate volumes from the corresponding organic solutions, in order to confer the specific lipid compositions required (as presented in Table 1), were added in a round bottomed flask; the total lipid concentration in the liposome dispersions was always equal to 20 mg/mL. The organic solvents were then evaporated by rotary evaporation (30 min; 105 rpm, 60 °C), and the lipids formed a thin lipid film on the sides of the flask. Any residual organic solvent was removed by subjection of the flask under a stream of N_2_ for 15 min. 

The resulting lipid film was subsequently hydrated at 60 °C with 1 mL of ammonium sulfate solution (120 mM, pH 5.5, 300 mOsm), if the vesicles were to be used for DOX loading, or with 1 mL calcein solution (100 mM, pH 7.40, 300 mOsm), if the vesicles were to be used for vesicle integrity studies, to form multilamellar vesicles (MLV). For physicochemical characterization, liposomes were hydrated with PBS buffer (pH 7.40, 300 mOsm). Vesicle size was then reduced by subjection to a high intensity (750 Watt) probe sonicator (Sonics and Materials, Leicestershire, UK) for (at least) two 15-min cycles, until the dispersion became completely clear, and then left at 60 °C for 1 h, to anneal any structural defects. Any traces of titanium, which may have leaked from the probe microtip, or any lipid aggregates present in the liposome dispersions, were removed by centrifugation at RCF 1600× *g* for 5 min (Scilogex 2012 microcentrifuge, Rocky Hill, CT, USA). 

The exact lipid content of the resulting liposomes was measured by the Stewart assay, a colorimetric method used routinely for the quantification of phospholipids [27]. Liposomes were purified from non-encapsulated solutes (calcein or DOX) by size exclusion chromatography (SEC), using a Sepharose 4B-CL column (40 × 1 cm), which was eluted with PBS buffer (pH 7.40), or by repeated ultracentrifugations for 1 h (each) at 60,000 rpm (Sorvall WX90 Ultra, Thermo Scientific, Waltham, MA, USA), depending on the need to re-concentrate the sample (or not) for the specific study that followed (if dilution occurring during SEC would cause a need for re-concentration, ultracentrifugation was preferred).

### 2.2. Physicochemical Properties of Liposomes

All the liposome types prepared were characterized for their lipid concentration, mean diameter, size distribution, and zeta-potential. For measurement of their size, the liposome dispersions were diluted to a final concentration of 0.4 mg/mL, and measured by dynamic light scattering (Malvern Instruments, Zetasizer Nano SZ, Malvern, UK), which enables the mass distribution of particle size to be obtained in the range between 0.3 nm–10 µm. Phosphate buffered saline (PBS 10 mM), pH 7.40 was used for dilution of LIP dispersions, after being filtered through polycarbonate filters (0.22 μm) (Millipore, UK). Particle size measurements were carried out with a fixed angle of 173° for backscatter correction, at 25 °C. The sizes reported correspond to the z-average means of the hydrodynamic diameters of the liposomes. 

For ζ-potential values, the electrophoretic mobility of the liposome dispersions was measured at 25 °C, by the same instrument. Zeta potential values were obtained (by the instrument) from the electrophoretic mobility, according to the Smoluchowski equation.

The percent incorporation of TREG in liposomes (compared to the initial amount of TREG added in the samples during liposome preparation), was quantified as reported before [18,19], in order to verify if the complete amount of TREG was indeed incorporated in the liposomes, and thus exclude any potential of micelle of small lipid aggregate formation. In brief, HPLC analysis of a specific quantity of liposomes (lipid amount) was carried out, before as well as after purification of the liposome dispersions, both by ultracentrifugation and size exclusion chromatography (as described above); integration of the corresponding peaks of TREG followed (see Appendix A). For this, the liposome samples were dissolved in MeOH and HPLC was performed with a Lichrosphere 100 RP-18 (5 mm) column; eluted with MeOH (as mobile phase) at a 1 mL/min flow rate, by a Shimatzu, LC-20AB Prominence Liquid Chromatography System. TREG elution was monitored at 330 nm. Blank liposomes (with no TREG) did not give any reading at the time period that TREG eluted from the column.

The physical stability of ARSL and TREG-ARSL was monitored by measuring their mean hydrodynamic diameter, polydispersity index, and zeta potential (as mentioned above) at specific time periods during their incubation at 4 °C, for up to 56 days.

### 2.3. Liposome Integrity Studies (Calcein Latency)

Calcein latency was monitored during incubation of the various liposome types in buffer as well as in the presence of serum proteins (80% FCS) as a measure of the liposome integrity in blood [28]. Calcein was encapsulated in the vesicles at a quenched concentration (100 mM) and for latency calculation, liposome samples (20 μL) were diluted with 4 mL buffer, pH 7.40, and the fluorescence intensity (FI) of the diluted samples was measured (EM 470 nm, EX 520 nm, 5 nm slits), before and after disruption of the liposomes by addition of Triton X-100 at a final concentration of 1% *v*/*v* (that ensures liposome disruption and release of all encapsulated dye). Percent latency (% *latency*) values were then calculated according to Equation (1): (1)$$\%\;\mathrm{Latency}=\frac{(1.1\cdot F_{AT})-F_{BT}}{1.1\cdot F_{AT}}\cdot 100$$
where, *F_BT_* and *F_AT_* are the calcein fluorescence intensities, before and after vesicle disruption by Triton X-100, respectively. *F_AT_* was multiplied with 1.1 for correction due to the dilution by Triton.

For evaluation of the specific disruptive effect that serum proteins cause towards liposome membranes, calcein *retention* (%) of vesicles was calculated from the *latency* of the liposomes during incubation in buffer, and the corresponding (at the same time point) *latency in FCS*, according to Equation (2):(2)$$\%\;\mathrm{Retention}=\frac{Latency-in-FCS}{Latency-in-buffer}\cdot 100$$

### 2.4. Loading/Release of DOX in/from Liposomes

DOX was loaded in LIP and TREG-LIP (control formulations) as well as in ARSL and TREG-ARSL by the active loading method, as described in detail elsewhere [16]. In brief, empty vesicles were pre-formulated in ammonium sulfate ([NH_4_]_2_SO_4_ (120 mM)), as described above. The vesicles were then ultra-centrifuged (Sorvall WX90 Ultra, Thermo Scientific) at 60,000 rpm for 1 h and re-suspended in PBS pH 7.4, for exchange of the dispersion media. Ultracentrifugation was used for external buffer exchange instead of size-exclusion chromatography, in order to avoid an extra step to re-concentrate the sample, since chromatography results in 3–5 times dilution of liposome dispersions. For the active loading of DOX, all types of (empty) liposomes, at a lipid concentration of 1.4 mg/mL (in PBS) were incubated in presence of a 0.2 mg/mL DOX solution (in PBS) (corresponding to a lipid/DOX ratio equal to 7:1 [*w*/*w*]), for various time periods (15, 30, 60 and 90 min) at 40 °C and 60 °C. After this, the liposomes were purified from non-encapsulated drug by two repeated ultracentrifugation steps (60,000 rpm for 1 h, each), in order to achieve high purification, without dilution of samples. Supernatants containing non-encapsulated DOX were collected, as well as the DOX-loaded liposome pellets, which were re-suspended in 1 mL of PBS, for measurement of DOX loading efficiency. For this, the liposome lipid concentration was measured by the Stewart assay, and the DOX concentration in free DOX solutions and liposomes was calculated by measuring the fluorescence intensity (FI, EX 485 nm, EM 590 nm, slits 5 nm) of DOX in every sample by a Shimadzu RF-Fluorescence Spectrophotometer. Liposomal samples were measured in the presence of Triton X-100 (1%) in order to dissolve the liposomes. To estimate the drug concentration, DOX calibration curves in the concentration range between 2.5 and 40 μg/mL were prepared, in media with similar composition as the samples (1% Triton X-100 in PBS, after ensuring that the presence of the dissolved lipids within the concentration range they were present in the samples, did not modify the calibration curve). 

The final D/L ratio (*w*/*w*) was estimated and compared with the initial one, for calculation of the DOX encapsulation efficiency (%) of each liposome type, according to Equation (3):(3)$$\mathrm{EE}\;(\%)=\frac{D/L_{final}}{D/L_{initial}}\cdot 100$$

The release of DOX from TREG-LIP, ARSL, and LIP (as control formulation) was studied by adding 0.5 mL of DOX-loaded liposome formulations at a lipid concentration of 1 mg/mL in dialysis tubing sacs (Servapor, with MW cutoff 14,000 Daltons) and placing the sacs in capped test tubes containing 10 mL of PBS buffer, pH 7.40. The test tubes were then closed and placed in a shaking incubator (Stuart Orbital Incubator) at 60 rpm and 37 °C for 48 h. At specified time points (0, 1, 2, h, 6, 18, 24, 48 h) 0.5 mL samples were taken from the buffer (volume was replaced with PBS) and DOX was quantified by measuring the sample FI, as described above. Sink conditions applied throughout the study.

### 2.5. Cell Culture Studies

Three types of cells were used in this study: (i) C57BL/6 mouse B16F10 skin melanoma cells (B16) (National Cancer Institute Tumor Repository, Frederick, MD, USA); (ii) Lewis lung carcinoma cells (LLC), and (iii) Human HEK-293 embryonic kidney cells (HEK) (American Type Culture Collection, Manassas, VA, USA), provided by Prof. G.T. Stathopoulos (Medical School, University of Patras). All cells were grown in RPMI 1640 medium supplemented with 10% FBS and 1% antibiotic-antimycotic solution (Invitrogen, Carlsbad, CA, USA). The cells were cultured at 37 °C, 5% CO_2_/saturated humidity. Medium was changed every 2–3 days. 

The toxicity of the various types of liposomes towards cancer and normal cells was evaluated by the corresponding reduction of cell viability after 24 h incubation with test (DOX-ARSL) or control formulations (DOX-solution, empty ARSL, and, DOX-solution—empty ARSL mixtures), by the MTT assay [29]. In more detail, for each experiment, cells were seeded overnight at 37 °C at a density of 3 × 10^4^ cells per well, in 24-well plates, until almost confluent, and then incubated for 24 h at 37 °C (5% CO_2_/saturated humidity) with 0.4 mL RPMI and 0.1 mL of the formulation evaluated in each case: (i) DOX-solution; (ii) DOX-ARSL; (iii) Empty ARSL (with the same final lipid concentrations with (ii)); and (iv) PBS (control). All formulations were pre-filtrated through a 0.22 μm Millipore filter. The medium/PBS (*v*/*v*) ratio was kept constant. Cell viability was determined by the MTT method, which is a colorimetric assay for assessment of the cell metabolic activity. For this, after the 24 h incubations with the cells, the medium was removed and the cells were washed three times with PBS before adding 0.5 mL of a 0.5 mg/mL solution of MTT in PBS. Two hours later, 0.5 mL of acidified isopropanol (0.33%) was added in each well, in order to disrupt the cells and solubilize the colored formazan crystals that formed. The optical density of the controls and samples was measured at 570 nm (Multiscan EX plate reader, Thermo). Viability (%) was calculated by the equation: (4)$$\mathrm{Viability}\;(\%)=\frac{{\mathrm{OD}-570}_{\mathrm{sample}}{\;-\;\mathrm{OD}-570}_{\mathrm{background}}}{{\mathrm{OD}-570}_{\mathrm{control}}{\;-\;\mathrm{OD}-570}_{\mathrm{background}}}\cdot 100$$
where, OD-570_control_ corresponds to untreated cells (or PBS control) and OD-570_background_ to MTT without cells. 1% Triton X-100 was used as a positive control of cytotoxicity, and resulted in viability values <5% for all cells.

For the cytotoxicity studies, the lipid/DOX ratios used during the loading of DOX into the liposomes were different from the ratio mentioned above (in Section 2.4), in order to achieve the specific DOX and ARSL concentrations required for induction of significant toxicity towards cancer cells. Test experiments were initially carried out for selection of the specific concentrations required, and finally the DOX concentrations selected were 1 µM and 3 µΜ, and the phospholipid concentrations of ARSL (empty or DOX-loaded) ranged between 0.5 µΜ and 50 µΜ. 

### 2.6. DOX Uptake by Cells

The uptake of DOX by B16 and LLC cells was measured. For this, cells were seeded overnight at 5 × 10^4^ cells/mL in RPMI at 37 °C (5% CO_2_/saturated humidity), and then incubated with DOX-liposomes or free DOX (control) for 3 h. DOX concentrations of 1 µM and 3 μM were used in the uptake experiments (after confirming that they are non-toxic towards the cells following a 3 h co-incubation period). After treatment of the cells with DOX-liposomes or DOX solution, the cells were washed twice with ice-cold PBS, and then detached and lysed by the addition of 3 mL of PBS and 2 mL of Triton X-100 (10%). The samples were collected and their fluorescence intensity was measured (EX 485 and EM 590 nm), for calculation of the DOX concentration from a calibration curve that was constructed by spiking known concentrations of DOX (between 0.25 µΜ and 4.5µM in similar media with that of the samples (Triton X-100 in PBS); the presence of lysed cells or liposomes did not modulate the FI of DOX control solutions (in the concentration range used). The auto-fluorescence of non-treated cells was also measured under identical conditions, and found to be very low (always below 0.1% of the FIs of the samples). The protein concentration in every sample was measured by the Bradford microassay. 

Finally, DOX uptake by cells was expressed as % Uptake (by comparison of the DOX measured in the cells and the total amount of DOX incubated with the cells), and also as nmoles DOX/μg protein. 

### 2.7. Transmission Electron Microscopy (TEM)

The morphology of ARSL and TREG-ARSL was studied by TEM after negative staining with 1% neutral phosphotungstic acid (PTA), washing twice with dH_2_O, and draining with a tip of a tissue paper. Samples were observed at 100.000 eV with JEOL (JEM-2100) TEM (Jeol, Tokyo, Japan)

### 2.8. Statistical Analysis

All results are expressed as mean ± SD from at least three independent experiments. The significance of variability between results from various groups was determined by two-way-ANOVA and individual differences between groups were tested by one-way ANOVA and Tukey’s multiple comparisons test. 

## 3. Results

### 3.1. Physicochemical Properties and Stability of ARSL and TREG-ARSL

All liposome dispersions were characterized for their size distribution (mean hydrodynamic diameter and polydispersity index), and their zeta-potential (Table 2). It should be pointed out that the liposomes characterized were formulated in PBS buffer, pH 7.40.

As seen in Table 2, the mean hydrodynamic diameter of all the liposomes prepared was in the nano-range, between 101 and 146 nm. The polydispersity index values were between 0.119 and 0.270 for all the liposomes, the lowest for LIP and the highest for TREG-LIP. All samples were monodisperse as seen in DLS reports shown in Appendix A (where analytical size distribution results for one sample of TREG-LIP and one sample of TREG-ARSL, before and after purification from any non-liposome incorporated components, are shown). ARSL, LIP, and TREG-LIP had mean diameters below 120 nm, while the TREG-ARSL were larger, having a mean diameter of 145.9 nm. In fact, the addition of TREG in liposomes was observed to confer an increase of vesicle mean diameter and PDI, compared to the corresponding liposomes without TREG; vesicle mean diameter was increased by 10% in the case of LIPs, and ~40% in the case of ARSLs, while PDI values were increased by 127% and 49%, respectively. All results are in good correlation with previously reported values [6,9,18,19,20]. 

Concerning the zeta-potential of the vesicles, values were always slightly negative, ranging between −2.51–−3.55, as anticipated by the facts that all the liposomes were pegylated and measured in saline-containing media, and the conventional liposome types LIP and TREG-LIP did not contain any charged lipid in their composition. Significantly lower zeta potential values (compared to ours), for vesicle with similar lipid compositions have been reported in the literature (between −11 ± 5 and −13 ± 7) [30]; however, those were measured in 10 mM HEPES, 300 mM sucrose, pH 7.0, while our measurements were carried out in the presence of high concentration of salts, which is known to result in substantial decreases of zeta potential values [31]. In order to verify the accuracy of our zeta potential measurements, we measured (using the same measurement conditions with all other samples) a control liposome formulation that contained a negatively charged lipid (phosphatidyl glycerol) in their membrane, and their z-potential value was −22 (Table 2). The current results for the ARSL are consistent with previous ones where PEGylated ARSLs were demonstrated to have very low negative zeta potential values when measured in saline-containing medium [9].

Quantification of the amount of TREG incorporated in TREG-LIP and TREG-ARSL by HPLC was carried out in the liposome dispersions, before and after purification from any material that was not incorporated in the liposomes. Representative HPLC chromatographs are shown in Appendix A. The results of this study verify the complete integration of TREG in the LIP and ARSL. Indeed, 95.8 ± 6.8% of the initial TREG was measured in TREG-LIP after purification by size exclusion chromatography. Furthermore, 100.20 ± 0.72% of the initial TREG was measured in TREG-ARSL after purification by size exclusion chromatography and 101.00 ± 3.23% after purification by ultracentifugation.

TEM studies show a similar vesicular morphology for ARSL and TREG-ARSL (Figure 1). 

The physical stability of ARSL and TREG-ARSL was monitored for a period of 56 d at 4 °C, and as seen in Figure 2, both liposome types were very stable for the full period evaluated; vesicle mean diameter, polydispersity, and zeta potential values were practically unmodified (no significant change was found) revealing the good stability of the formulations. 

### 3.2. Calcein Latency and Retention

The percent Latency of vesicle-encapsulated calcein during incubation of the vesicles in PBS buffer and FCS was evaluated for all the liposome types. As seen in Figure 3A, the control liposomes (LIP) (without arsonolipids) preserved high calcein latency during incubation in PBS; however, they demonstrated a gradual linear release of vesicle-entrapped calcein when they were incubated in FCS. When TREG was added in the liposomes (TREG-LIP), the calcein latency dropped by about 7 percent during the first hour of incubation, but after that, the decrease was minimal. The initial drop in calcein latency was higher during incubation in PBS compared to FCS (Figure 3B). Regardless of the initial decrease of calcein latency in the first hour, the calcein leakage from TREG-LIP during incubation in FCS (Figure 3B) was much slower compared with that from LIP (Figure 3A), during the period between 1–72 h, indicating that the addition of TREG in the liposomes rendered the liposomes leakier in buffer but increased their integrity in the presence of serum proteins. 

A similar effect of TREG on the integrity of vesicle during incubation in buffer and FCS was also demonstrated in the case of ARSL (Figure 3C,D). However, since the specific ARSL composition used herein has high integrity during incubation in FCS (compared to conventional LIP) on its own (Figure 3A), the stabilizing effect of TREG on these vesicles (ARSL) towards their disruption by serum proteins was less pronounced. 

The retention of calcein in vesicles is a better measure (compared to calcein latency) of the disruptive effect that serum proteins demonstrate towards membranes, since any leakage of the vesicle-entrapped dye (calcein) that is not caused by disruption of the membrane due to the presence of serum proteins or by protein interactions with membrane components, is omitted. Thereby, the stabilizing effect of TREG on LIP and ARSL towards protein-induced disruption is better seen in Figure 4, where calcein retention values are plotted. Indeed, the profound stabilization of plain LIP towards serum proteins, by the addition of TREG in their membranes (TREG-LIP) is proved, as well as the (lower) effect realized when TREG is incorporated in ARSL (TREG-ARSL) (Figure 4). 

### 3.3. DOX–Loading Efficiency and Release Studies

The active loading method was applied for DOX loading into ARSL and TREG-ARSL. LIP and TREG-LIP were also studied under identical conditions as control formulations for comparison.

As seen in Figure 5, the loading of DOX achieved in all liposome types and loading protocols applied ranged between 23% and 92% when the incubation of DOX with empty vesicles was carried out at 40 °C, while loading was significantly higher, ranging between 55% and 98%, when the incubation was done at 60 °C. The effect of the temperature on DOX loading could be related with the high phase transition temperatures of the lipid components of the liposomes, which is 55 °C for DSPC and >60 °C for C16 arsonolipid [25], and probably lower for TREG, which contains C_16_ fatty acids linked to the glycerol backbone, resembling DPPC (with 41 °C transition temperature). When the DOX/vesicle incubation was carried out at the lowest temperature (40 °C), the duration of incubation had a significant effect on the amount of drug loaded, since DOX-loading increased significantly between 30 and 90 min of incubation for all vesicle types studied. TREG-ARSL were an exception to the latter result, since their loading already reached the maximum value at 30 min. TREG-ARSL loading values were the highest, compared to all the other vesicle types (*p* < 0.001). In fact, the addition of TREG in LIP as well as in ARSL significantly increased (*p* < 0.001) the loading of DOX in the corresponding vesicles for all the incubation time periods tested at 40 °C, but only for the 15 min and 30 min periods when the incubation was carried out at 60 °C. At 60 °C, maximum loadings in all liposome types were achieved faster. Another interesting point is that DOX encapsulation in ARSL and TREG-ARSL at 40 °C was always (for all incubation periods) higher than the corresponding loading values (at the same conditions) in LIP and ARSL, respectively, suggesting that the presence of TREG or arsonolipid in the membrane of liposomes rendered them more accessible for DOX loading at lower temperature. This positive effect of TREG and arsonolipid on the loading efficiency of the vesicles with DOX may be attributed to modulations in the liquid ordered (LO) and liquid disordered transition of the membranes, or chemical interactions (e.g., hydrogen bonding or dipole-dipole interactions) between specific groups of each molecule (the polar head group of TREG and/or arsonolipid with DOX); however, more studies are required in order to prove any of these suggestions. 

The release of DOX from some liposome types was also studied in order to see if the presence of TREG in the liposomes, or if the structure of ARSL would result in faster leakage of DOX from such vesicle types. As seen in Figure 6, the release of DOX was faster in the first 6 h of incubation, by which approx. 18% of the encapsulated DOX was released; after that, a slower release of the drug was demonstrated up to about 25% at 48 h. The non-released amounts of DOX from all the liposome samples were found in the dialysis sacs after the 48 h period was completed. A similar profile of DOX release was observed for all the liposomes, indicating that TREG or arsonolipids do not modify the capability of the liposomes to retain DOX, at least when DOX is remotely loaded into the vesicles.

### 3.4. In Vitro Anticancer Activity—Preliminary Results

Concerning the cytotoxicity of DOX-loaded ARSL, preliminary experiments were carried out with two cancer cell lines, B16 and LLC cells, and one normal cell line, HEK cells. Control experiments with empty ARSL and free DOX were initially carried out in order to identify the concentrations that induced significant toxicities towards the two cancer cell lines, and then the appropriate D/L ratios were used for preparation of the DOX-ARSL. As seen in Figure 7, the B16 cells are more resistant to ARSL (concentration range 0.38–2.4 mM (of corresponding phospholipids)) having viability values between 85% and 49% (Figure 7A), compared to the LLC cells which at similar ARSL concentrations (0.44–2.64 mM (of corresponding phospholipids)) demonstrated viabilities between 64% and 12.5% (Figure 7B). HEK cells were even more resistant than B16 cells to ARSL (Figure 7C), at most concentrations studied (0.44–1.45 mM of corresponding phospholipids), in accordance with previous reports that demonstrated higher toxicity of ARSL towards cancer cells, compared to normal cells [4,5,6]. An exception was observed at the highest ARSL concentration used (2.4 mM), at which the viability of HEK cells was drastically decreased to 20.2%, and was lower than the corresponding viability of B16 cells. 

Concerning the effect of free DOX (solution) on the cells, at 1 μM DOX, viability values of 53.4 ± 7.9%, 64.4 ± 2.4%, and 53.6 ± 1.7% were measured for B16, LLC, and HEK cells, respectively. When 3 times higher DOX (concentration) was used, the corresponding viability values for B16, LLC, and HEK were 47.7 ± 6.1%, 36.1 ± 4.8% and 45.2 ± 8.9% respectively, indicating that the B16 and HEK cells were more resistant to DOX (as also demonstrated for empty ARSL) compared to LLC cells; indeed, the viabilities of B16 and HEK cells were not significantly reduced (*p* > 0.05) when the DOX concentration was increased by 3-fold, oppositely to LLC cells that demonstrated a 56% reduction in viability (*p* < 0.0001).

Interestingly, when the LLC cells were incubated with DOX-ARSL (Figure 7B, light gray bars), in most of the cases (the three last DOX/arsonolipid combinations tested), the viability values were significantly lower (*p* < 0.001) compared to the corresponding viability value attained with the same concentration of free DOX (black bars), and also compared to the empty ARSL (patterned bars), with the exception of the last combination at which 2.64 mM of empty ARSL conferred very high toxicity towards LLC cells (12% viability). Even more interesting is the fact that in one case (1 μM DOX + 0.88 mM lipid) the reduction in LLC cell viability induced by DOX-ARSL (Figure 7B, light gray bars) was significantly higher compared to the effect of the corresponding free-DOX + empty ARSL mixture (dark gray bars). A marginal (but not significant) difference between the cytotoxic effect of DOX-ARSL and the corresponding mixture (towards LLC cells) was also observed with the last combination, (DOX2/ARS4, (3 μM DOX + 2.64 mM lipid) shown in Figure 7B.

In the more resistant B16 cells (Figure 7A), the effect of DOX-ARSL was almost never significantly higher compared to that of the same concentration of free-DOX, with the exception of the last combination tested (3 μM DOX + 2.4 mM lipid). In the latter case, the DOX-ARSL was again (as also mentioned above in one case for LLC cells) significantly more toxic compared to the corresponding mixture of free-DOX and empty-ARSL. In fact, the DOX-ARSL types were always as toxic or significantly more toxic towards B16 cells, compared to the corresponding mixtures of (free) DOX + (empty) ARSL. 

Interestingly, DOX-ARSL were demonstrated to be significantly less toxic towards (normal) HEK cells, compared to their corresponding mixtures (at all ARSL and DOX combinations applied), indicating an important advantage of the novel formulations. Another very interesting observation (from a therapeutic point of view) is that DOX-ARSL (again at all the concentrations tested) were less toxic towards HEK cells, compared to the two cancer cells.

### 3.5. Cellular Uptake of DOX

The uptake of DOX as well as DOX-ARSL by B16 and LLC cells after 3 h of co-incubation of the cells with free-DOX (as solution) are presented in Table 3. As seen, both types of cells take up substantially higher amounts of DOX when the DOX is free compared to when it is encapsulated in ARSL (*p* < 0.001 in both cases). In addition, although the uptake values for free DOX are similar for both cell types, the uptake of DOX from DOX-ARSL is significantly higher by LLC cells, compared to B16 cells (*p* < 0.01), which is consistent with the higher toxicity of DOX-ARSL towards LLC cells, compared to B16 cells. 

## 4. Discussion

Herein, we investigated the potential to incorporate TREG (a previously synthesized and studied lipidic derivative of curcumin) [18] in ARSL and its effect on ARSL stability. Furthermore, we evaluated the ability to load ARSL and TREG-ARSL with DOX using the remote liposome loading method, in order to develop potential anticancer chemotherapeutic systems that will combine the anticancer activities of ARSL and DOX, together with the potentially enhanced pharmacokinetics and targeting capability provided by TREG. 

TREG-LIP and TREG-ARSL, as well as LIP and ARSL (which were prepared as control vesicles), had mean diameters in the nano-range, between 101 and 145 nm (Table 2), polydispersity values below <0.270, and slightly negative zeta potential values. The low zeta potential value measured for ARSL despite the highly negative charge of arsonolipids, was attributed to the fact that the particular ARSL were PEGylated, and was consistent with previous results [9]. For the liposomes with no arsonolipids in their composition, the differences with other reported values [30] were attributed to the different measurement conditions used [31]. The complete incorporation of TREG in the liposome membranes was verified by HPLC measurements that were carried out before and after purification of the lipid dispersions from non-liposome associated materials (by size exclusion chromatography) (Appendix A), ruling out any suspicion that TREG micelles or aggregates could be present in the formulations, as also evident in the analytical size reports of DLS measurements (Appendix A), and their morphology (Figure 1).

After confirming the capability to formulate stable nanosized TREG-ARSL that incorporate the complete amount of TREG added in the lipid phase during their preparation, and demonstrate high physical stability during storage at 4 °C (Figure 2), their integrity in the presence of serum proteins was evaluated. The current results demonstrated that the incorporation of TREG in ARSL had a marginal effect to increase their stability towards any disruptive actions of serum proteins, and confirmed previous findings about the stabilizing (towards disruption by serum proteins) effect of TREG on TREG-LIP [20,21]. 

Concerning the capability of using the active loading protocol [15,16] to load DOX into ARSL and TREG-ARSL, the current results (Figure 5) confirmed for the first time that high amounts of DOX can be loaded into ARSL, opening the path for exploration of potential synergistic anticancer effects of the two compounds when co-existing in a vesicular format. Additionally, it was proven that when liposomes or ARSL incorporate TREG, this does not prevent in any way their capability to be remotely loaded with DOX. On the contrary, high DOX loadings were achieved faster in TREG-LIP and ARSL compared to the corresponding control vesicles. Whether the latter results are attributed to modulations in the liquid ordered (LO) and liquid disordered transition of the membranes, or to the decrease of the hydrophilic properties of the vesicle surface that may result in faster attraction of DOX molecules into the lipid membrane, or perhaps to chemical interactions (e.g., hydrogen bonding or dipole-dipole interactions) between specific groups of the TREG and/or the arsonolipid with DOX, we cannot be sure. Importantly, the release profile of DOX from all the vesicle types was found to be similar, proving that TREG or arsonolipids do not cause faster (compared with DOX-LIP) leakage of DOX (Figure 6).

The in vitro experiments carried out with DOX-ARSL on B16, LLC and HEK cells demonstrated interesting anticancer potential of DOX-ARSL toward both cancer cell types at specific DOX and ARSL combinations (Figure 7). In most cases, the cytotoxic effect of DOX-ARSL towards cancer cells was higher than that demonstrated when the corresponding mixtures of free DOX and empty ARSL were used, indicating the importance of combining the two compounds in the same vesicular structure. Oppositely, the situation was different for HEK cells, since DOX-ARSL (at all concentrations) were found to be less toxic compared to the corresponding mixtures. The previous finding, together with the fact that in all cases DOX-ARSL were less toxic towards HEK cells and significantly more toxic towards the cancer cells (the less resistant LLC cells as well as the more resistant B16 cells) indicate some very strong advantages of using the particular formulations as anticancer therapeutics. 

The uptake of free DOX by both cell types is substantially higher compared to that of ARSL-entrapped DOX (Table 3), suggesting that the cytotoxic effect of the comparatively low amounts of DOX taken-up by the cells (when they are incubated with DOX-ARSL) is highly enhanced when appropriate amounts of arsonolipids are taken up at the same time (in the form of DOX-ARSL). 

In order to understand if the current results indicate synergism, additive effect or antagonism between DOX and ARSL, we calculated the corresponding combination index (CI), for each case, according to the previously reported equation: CI = (D1/Dm1) + (D2/Dm2) [22,32]. Here, D1 and D2 represent the dose of the drugs in combined administration (DOX-ARSL) and Dm1 and Dm2 represent the dose required to produce the same effect when used alone (free DOX or empty ARSL). It has been reported that when CI < 1, synergism is indicated; when CI = 1, additive nature is indicated; and when CI > 1, antagonism is indicated [32,33]. The CI values that could be calculated from the current results are reported in Table 4. The values for the free DOX and empty ARSL (Dm1 and Dm2) were taken from the corresponding results of cell viability in presence of increasing concentrations of free compounds (Appendix A). For the B16 cells, some values for the free compounds were not available, since even at very high concentrations of empty ARSL, the viability of B16 cells was never below 45–50%; therefore, some CI values could not be estimated. As seen in Table 4, in two cases for B16 cells and in one case for LLC cells, the CI values indicate synergism between arsonolipids and DOX; while in other cases in LLC cells, antagonism is indicated. 

The current preliminary in vitro studies are not sufficient to fully explain the mechanisms of the relative cytotoxic effects demonstrated by DOX-ARSL towards cancer and normal cells. Nevertheless, the highly interesting current finding in regards to the cytotoxic properties of DOX-ARSL towards cancer cells and their reduced toxicity towards normal (HEK) cells fully justify further in vivo exploitation of this particular combination therapy nano-system. Furthermore, the fact that the particular (DOX-loaded) ARSL can also incorporate TREG in their membranes, with a slight improvement in their in vitro integrity during incubation in the presence of proteins, unlocks a series of potential applications for difficult-to-treat pathologies, such as brain tumors, that should be additionally investigated in future projects. In fact, the additive or synergistic anticancer activity of DOX/curcumin combinations suggested by several recent studies [34,35,36,37] provide an extra reason to further investigate the anticancer activity of DOX-loaded TREG-ARSL. 

## 5. Conclusions

The current results prove the potential anticancer properties of the novel types of ARSL, and justify their future exploitation as anticancer therapeutic systems.

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
