# Peer review of "Preparation, Physicochemical Properties, and In Vitro Toxicity towards Cancer Cells of Novel Types of Arsonoliposomes"

_pharmaceutics, 2020, doi:10.3390/pharmaceutics12040327_

Round 1
Reviewer 1 Report
The authors have addressed all my comments properly. My decision is to accept the revised manuscript to publish in the Pharmaceutics Journal.
Author Response
We thank this reviewer for finding our article suitable for publication in Pharmaceutics
Reviewer 2 Report
The authors adequately addressed and clarified all the raised issues and questions.
One minor comment: For the description of the centrifugation protocol, RCF (or x g) should be given instead of RPM. (RPM with the type of the rotor would be also sufficient, but rpm alone is not)
Author Response
We thank this reviewer for finding our answers and actions taken adequate to adreess all the previous conserns. We agree that the RCF is a better description for the centrifugal force applied and made the appropriate revision in the manuscript (see methods section lines 148-149)
Reviewer 3 Report
I am happy with the replies the authors made. The MS is much better now and I suggest its publication. I only recommend the authors to present their new TEM images in the main part of the paper, not as supporting info. In fact, the images are quite good, the authors underestimate these results. I feel that these images will look good in the main paper.
Author Response
We would like to thank this reviewer for finding our article much better now and for suggesting publication. We are also grateful to this reviewer for finding that the TEM images could be added in the main part of the article. Thereby, we revised the article accordingly and placed the TEM image figure, as Figure 1 in the revised ms, and made all the required changes becasue of this. We thank this reviewer for the very interesting critisism and points raised, that helped us to make our article better.
This manuscript is a resubmission of an earlier submission. The following is a list of the peer review reports and author responses from that submission.
Round 1
Reviewer 1 Report
The work is well written and discussed. No modifications are needed.
Reviewer 2 Report
Decision: Minor Revision
Comments:
- The manuscript is having title as “Preparation and physicochemical properties of curcumin-coated and doxorubicin loaded arsonoliposomes: Preliminary anticancer activity” which seems confusing and authors are suggested to reframe the title. Here, it says curcumin coated, when something is coated, that means mostly the outer surface is coated with the material however here curcumin is embedded in lipid membrane. Hence, the concept and title are not matching.
- Page 3, line, 99, authors may include following or similar reference to cite thin film hydration method;
Priyanka Bhatt et al. Liposomes encapsulating native and cyclodextrin enclosed Paclitaxel: Enhanced loading efficiency and its pharmacokinetic evaluation, International Journal of Pharmaceutics 536 (2018) 95-107, PMID: 29175440
- Authors should include parameters applied in preparation of liposomes in method such as, evaporation time, rotation and temperature to prepare film, hydration volume, probe sonication parameters etc, to help other researchers to get reproducible results.
- Authors have used very high rpm for centrifugation and separation of titanium particles after sonication. Is that much high rpm is required to settle down titanium particles? What are the chances of liposomes to settle down at this high rpm? Please justify.
- Fig 5A, bar for mixture in DOX1/ARS1 is missing. Please justify.
Reviewer 3 Report
In their paper, Zagana et al. describe the preparation and in vitro testing of a liposomal drug delivery system composed of an arsonolipid (ARS) and a lipid derivative of curcumin (TREG) loaded with doxorubicin. Though the paper is technically sound and builds on the previous publications of the authors in the topics of arsonolipids and curcumin-lipid derivatives, unfortunately the reviewer feels that the study has serious flaws, and some experiments are not conducted correctly.
- Preparation of liposomes. Preparation of unilamellar liposomes with probe sonication is an old and – to some extent – outdated procedure. Nowadays, vast majority of studies dealing with liposomes utilize the extrusion method for liposome preparation. Removal non-encapsulated DOX with ultracentrifugation is also outdated, the use of gel filtration is recommended.
- Zeta potential. The reported Zeta-potential values indicate practically no surface charge. DOXIL (marketed PEGylated liposomal doxorubicin since 1995) has a Zeta potential of -23 mV to -25 mV. Since the LIP sample has very similar composition to DOXIL, this contradiction most probably caused by misconducted experiment or liposome preparation. DOXIL (Caelyx in Europe) is readily available, so the authors are suggested to measure Caelyx as a control for their analytical methods.
- Size distribution by DLS. DLS is widely used for the size characterization of monodisperse unilamellar liposomes. But DLS is not applicable for heterogeneous systems, e.g. if micelles are present besides the liposomes. Since probe sonication was used for preparation, the formation of lipid aggregates smaller than the liposomes cannot be ruled out. This is especially critical for TREG-LIP. Morphological characterization by cryo-TEM or AFM is necessary in such case.
- Calcein latency study. The definition of Latency is clear from the description, only the multiplication factor of 1.1 should be explained (i.e. due to dilution, but this is not stated explicitly in the text). Latency measurements shows that TREG-LIP and ARSL already leaking calcein in PBS resulting in latency around 80%. Control liposomes are stable in PBS and leaks in FCS as expected (because FCS contain albumin and lipoproteins which can bind lipids from the liposomes). TREG-LIP and ARSL leaks in FCS very similarly as in PBS. For an outsider reader, this shows that TREG-LIP and ARSL are less stable than the control LIP. The authors introduce ‘Retention’ which is the calcein leaking in FCS normalized to that in PBS. This is misleading, because TREG-LIP and ARSL are leaking already in PBS! If this is the true interpretation of these results, the conclusion about the stabilizing effect of TREG is incorrect!
- Doxorubicin loading. In general, the DOX loading efficiencies are too low for control LIP. Since LIP contains DSPC, which has a transition temperature at 55 C, loading at 40 C will be insufficient for sure (data support this). ARS and TREG contain dipalmitoyl lipids (diC16) which has lower transition temperature (see the literature on thermosensitive liposomes), this, and the ‘leaky’ nature of ARSL and TREG-LIP might explain the higher loading efficiency. This should be discussed in more details, and a measurement of the liquid ordered (LO) and liquid disordered transition (eg. the change of CH2 vibration frequency by FTIR) should be performed. DOX release studies should be also performed to reveal the stability of the final liposomal samples.
Reviewer 4 Report
The paper by Zagana et al is not recommended for publication for the following reasons:
- Liposomes characterisation is poor. The authors present no TEM data. I doubt that a suspension with such a low zeta potential will be that stable (up to ca 60 dyas!), as the authors claim. I do not also believe in the reported results on size distribution, this has to be confirmed using TEM.
- Anticancer activity cannot be established based purely on in vitro test. Especially when the authors use only cancer cells. The authors may refer to cytotoxicity, which by no means represents anticancer activity. This part of the study is particularly poor. The authors, for example, do not show any cellular uptake, cell morphology, etc. They have used only MTT test, which is not sufficient even for cytotoxicity study. I fail to see control cells results in the Figure 5.
- The experimental section is not written well. For example, I do not see the details on cellular fluorescence detection.
- The authors claim in the introduction that curcumin exerts anticancer activity. This is not true. The authors cite a paper in a dodgy journal, whereas a solid body of evidence exists which defies any "anticancer" properties of curcumin (see Nelson et al, J. Med. Chem. 2017, 60, 5, 1620-1637)
- The paper is written poorly, with numerous typos. In particular, I could not understand which country the authors are from. Hellas? Is this Greece? If the address is given in English, then why not using the English word for this country?